# Reasoning with Language Model is Planning with World Model

**Shibo Hao**[*♣]  **Yi Gu**[*♣]  **Haodi Ma**[◇]  **Joshua Jiahua Hong**[♣]
**Zhen Wang**[♣♠]  **Daisy Zhe Wang**[◇]  **Zhiting Hu**[♣]

♣UC San Diego, ◇University of Florida
♠Mohamed bin Zayed University of Artificial Intelligence
{s5hao, yig025, jjhong, zhw085, zhh019}@ucsd.edu
{ma.haodi, daisyw}@ufl.edu

## Abstract

Large language models (LLMs) have shown remarkable reasoning capabilities, particularly with chain-of-thought (CoT) prompting. However, LLMs sometimes still struggle with problems that are easy for humans, such as generating action plans to achieve given goals in an environment, or performing complex math or logical reasoning. The deficiency stems from the key fact that LLMs lack an internal *world model* to predict the world *state* (e.g., environment status, intermediate variable values) and simulate long-term outcomes of actions. This prevents LLMs from performing deliberate planning akin to human brains, which involves exploring alternative reasoning paths, anticipating future states and rewards, and iteratively refining existing reasoning steps. To overcome the limitations, we propose a new LLM reasoning framework, **Reasoning via Planning (RAP)**. RAP repurposes the LLM as both a world model and a reasoning agent, and incorporates a principled planning algorithm based on Monte Carlo Tree Search for strategic exploration in the vast reasoning space. During reasoning, the LLM (as agent) incrementally builds a reasoning tree under the guidance of the LLM (as world model) and rewards, and efficiently obtains a high-reward reasoning path with a proper balance between exploration *vs.* exploitation. We apply RAP to various challenging reasoning problems including plan generation, math reasoning, and logical inference, and demonstrate its superiority over strong baselines. RAP with LLaMA-33B even surpasses CoT with GPT-4, achieving 33% relative improvement in a plan generation setting.[1]

## 1 Introduction

Large language models (LLMs) have exhibited emergent reasoning abilities in a wide range of tasks (Brown et al., 2020; Chowdhery et al., 2022;

---

*equal contribution
[1]The code is available at https://github.com/Ber666/llm-reasoners

OpenAI, 2023). Recent approaches further boost their ability by prompting LLMs to generate intermediate reasoning steps, e.g., Chain-of-Thought, CoT (Wei et al., 2022) or answer a series of sub-questions, e.g., least-to-most prompting (Zhou et al., 2022). However, LLMs still face difficulties with tasks that humans find easy. For example, in creating action plans to move blocks to a target state, GPT-3 (Brown et al., 2020) achieves a success rate of only 1%, compared to 78% for humans (Valmeekam et al., 2022); these models also struggle with complex tasks that require multiple steps of math, logical, or commonsense reasoning (Huang and Chang, 2022; Mialon et al., 2023).

Humans possess an internal **world model**, a mental representation of the environment (Johnson-Laird, 1983, 2010; Gentner and Stevens, 2014), which enables humans to simulate actions and their effects on the world's state for deliberate **planning** for complex tasks of motor control, imagery, inference, and decision making (Tolman, 1948; Briscoe, 2011; Schulkin, 2012; LeCun, 2022). For example, to make an action plan towards a goal, planning with the world model involves exploring various alternative courses of actions, assessing the likely outcomes by rolling out possible future scenarios, and iteratively refining the plan based on the assessment (Huys et al., 2012; Gasparski and Orel, 2014; Ho et al., 2021). This is in stark contrast to the current LLM reasoning, which instinctively generates a reasoning trace in an autoregressive manner. In particular, we identify several key limitations of the current reasoning with LLMs, including **(1)** the lack of an internal world model to simulate the *state* of the world (e.g., the configuration of blocks, the values of intermediate variables), which is the foundation of human planning; **(2)** the absence of a *reward* mechanism to assess and guide the reasoning towards the desired state; and due to both limitations, **(3)** the incapability of balancing *exploration vs. exploitation* to efficiently explore vast

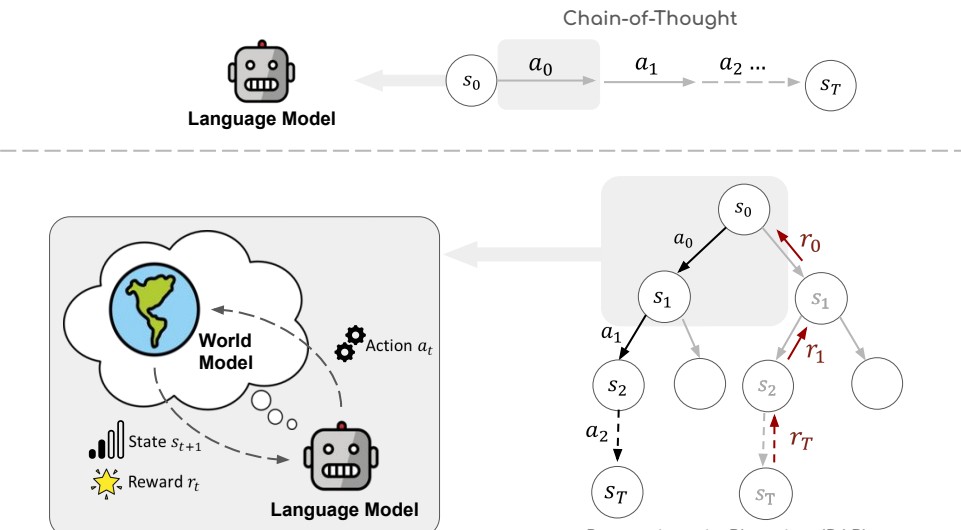

Figure 1: An overview of Reasoning via Planning (RAP). Compared with previous LLM reasoning methods like Chain-of-Thought (Wei et al., 2022), we explicitly model the world state from a world model (repurposed from the language model), and leverage advanced planning algorithms to solve the reasoning problems.

reasoning space.

To address these limitations, this paper proposes a new framework, **Reasoning via Planning (RAP)**, that enables LLMs to reason in a manner close to humans' conscious planning. RAP augments the LLM with a world model, and reasons with principled planning (specifically *Monte Carlo Tree Search, MCTS*) to produce high-reward reasoning traces after efficient exploration (Figure 1). Notably, we acquire the world model by repurposing the LLM itself with appropriate prompts. During the reasoning, the LLM strategically builds a reasoning tree by iteratively considering the most promising reasoning steps (*actions*) and using the world model (the same, repurposed LLM) to look ahead for future outcomes. The estimated future rewards are then backpropagated to update the LLM's beliefs about the current reasoning steps, guiding it to refine the reasoning by exploring better alternatives. Our MCTS-based planning effectively maintains a proper balance between exploration (of unvisited reasoning traces) and exploitation (of the best reasoning steps identified so far).

We show RAP is a general framework applicable to a diverse range of challenging problems and achieves substantial improvements over recent popular LLM reasoning methods. For plan generation, particularly in 2/4/6-step problems of Blocksworld (Valmeekam et al., 2023), RAP achieves an average success rate of 64% while CoT fails almost completely. Moreover, LLaMA-33B with RAP surpasses GPT-4 with CoT by 33% relative improvement. In the domains of mathematical reason-

ing, such as GSM8K (Cobbe et al., 2021) and logical inference exemplified by PrOntoQA (Saparov and He, 2022), RAP also consistently improves over strong baselines, including CoT, least-to-most prompting, and their self-consistency variants.

## 2 Related Work

**Reasoning with LLMs.** LLM reasoning typically involves decomposing complex questions into sequential intermediate steps (a.k.a. chains) before producing the final answer, exemplified by Chain-of-Thought (CoT) prompting and its variants (Wei et al., 2022; Kojima et al., 2022). The basic CoT generates chains all at once and can induce additional errors as the step count increases. Self-Consistency (Wang et al., 2022) samples multiple chains to choose the best answer via majority voting. Least-to-most prompting (Zhou et al., 2022) reduces the question into simpler subquestions and answers them sequentially. Similar to our reward formulation, recent works have explored self-evaluation approaches to provide feedback for intermediate steps (Welleck et al., 2022; Shinn et al., 2023; Paul et al., 2023). Aligned with our state formulation, Li et al. (2022) incorporate latent "situations" into LLMs, referring to the state of entities from the context. More relevantly, recent works have started to explore more complex structures guided by some search algorithms. For instance, CoRe (Zhu et al., 2022) fine-tunes reasoning step generator and verifier for math word problems with MCTS for decoding. Concurrently to our work, Yao et al. (2023) apply heuristic-based

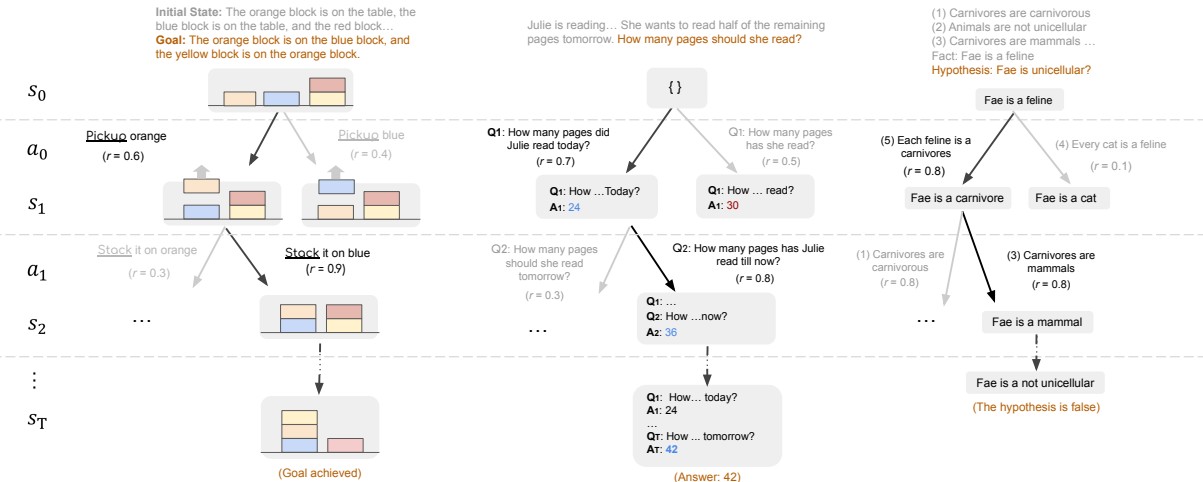

Figure 2: RAP for plan generation in Blocksworld (left), math reasoning in GSM8K (middle), and logical reasoning in PrOntoQA (right).

search, like depth-/breadth-first search, for better reasoning paths. However, none of the above methods formally introduce the world model and instantiates the reward and state into a unified framework. Compared with these search-guided methods, RAP is a more principled framework to combine world model and reward with advanced planning.

**Planning with LLMs.** Planning, a central ability in intelligent agents, involves generating a series of actions to achieve a specific goal (McCarthy, 1963; Bylander, 1994). Classical planning methods have been widely adopted in robots and embodied environments (Camacho and Alba, 2013; Jiang et al., 2019). Recently, prompting LLMs to do planning directly has gained attention and shown potential (Huang et al., 2022; Singh et al., 2022; Ding et al., 2023). Moreover, based on LLMs' powerful programming ability (Lyu et al., 2023; Jojic et al., 2023; Liu et al., 2023), recent works first translate natural language instructions into the executable programming languages, such as Planning Domain Description Language (PDDL), and runs classical planning algorithms, such as LLM+P (Liu et al., 2023). However, code-based planning is constrained by its narrow domains and the environment, while RAP can handle open-domain problems, such as math and logical reasoning. More related works on *world models and planning* are discussed in the Appendix D.

## 3 Reasoning via Planning (RAP)

In this section, we present the Reasoning via Planning (RAP) framework that enables LLMs to strategically plan a coherent reasoning trace for solving a wide range of reasoning tasks. We first build the world model by repurposing the LLM with

prompting (Section 3.1). The world model serves as the foundation for deliberate planning, by allowing the LLM to plan ahead and seek out the expected outcomes in the future. We then introduce the rewards for assessing each state during reasoning in Section 3.2. Guided by the world model and rewards, the planning with Monte Carlo Tree Search (MCTS) efficiently explores the vast reasoning space and finds optimal reasoning traces (Section 3.3). Finally, when multiple promising reasoning traces are acquired during planning, we further introduce an aggregation method in Section 3.4 that yields an ensembled result and further boosts the reasoning performance.

### 3.1 Language Model as World Model

In general, a world model predicts the next *state* of the reasoning after applying an *action* to the current state (Ha and Schmidhuber, 2018b; Matsuo et al., 2022). RAP enables us to instantiate the general concepts of state and action in different ways depending on the specific reasoning problems at hand (Figure 2). For example, in Blocksworld, it is natural to define a state as the configuration of blocks (described in natural language), and an action to be a behavior of moving a block (e.g., "pickup the orange block"). In a math reasoning problem, we use the state to represent the values of intermediate variables, and set an action to be a subquestion that drives the reasoning to derive new values. In logical reasoning, a state is a fact we are focusing on, and an action is to choose a rule for the next deduction.

With the definition of state and action, the reasoning process can thus be described as a Markov decision process (MDP): given the current state

$s_{t,t=0,1,...,T}$, e.g., the initial state $s_0$, the LLM (as a reasoning agent) generates an action space by sampling from its generative distribution $a_t \sim p(a|s_t, c)$, where $c$ is a proper prompt (e.g., in-context demonstrations). Once an action is chosen, the world model then predicts the next state $s_{t+1}$ of the reasoning. Specifically, we repurpose the *same* LLM to obtain a state transition distribution $p(s_{t+1}|s_t, a_t, c')$, where $c'$ is another prompt to guide the LLM to generate a state. For instance, in Blocksworld, the LLM (as the world model) generates text $s_{t+1}$ to describe the new configuration of blocks, given previous state $s_t$ and the action $a_t$.

Continuing the process results in a reasoning trace, which consists of a sequence of interleaved states and actions $(s_0, a_0, s_1, \ldots, a_{T-1}, s_T)$. This differs from the previous reasoning methods, such as Chain-of-Thought (Wei et al., 2022), where the intermediate reasoning steps consist of only a sequence of actions, e.g., ($a_0$ = "pickup red block", $a_1$ = "stack on yellow block", ...) (see comparisons in Figure 1). Augmenting the reasoning with the (predicted) world states helps the LLM with a more grounded and coherent inference. Note that the full reasoning trace is simulated by the LLM itself (as a reasoning agent with an *internal* world model) without interacting with the *external* real environment. This resembles humans contemplating a possible plan in their minds. The capability of simulating future states, by introducing the world model, allows us to incorporate principled planning algorithms to efficiently explore the vast reasoning space as described in Section 3.3.

### 3.2 Reward Design

During reasoning, we want to assess the feasibility and desirability of each reasoning step, and guide the reasoning based on the assessment (Section 3.3). The assessment of each reasoning step (i.e., applying an action $a_t$ to the state $s_t$) is performed by a *reward* function $r_t = r(s_t, a_t) \in \mathbb{R}$. Similar to the state and action, the reward function can be specified in different ways to accommodate any knowledge or preferences about the reasoning problem of interest. Here we introduce several common rewards applicable to different tasks and shown to be effective in our experiments.

**Likelihood of the action.** When an action is generated by the LLM conditioning on the in-context demonstration and the current state, the probability of the specific action reflects the LLM's preference.

We thus can incorporate the log probability of the action as a reward. This reward reflects the "instinct" of LLMs as an agent, and can be also used as a prior for which action to explore.

**Confidence of the state.** State prediction is non-trivial in some problems, e.g., in math reasoning (Figure 2, middle), given an action (i.e., a subquestion), the world model predicts the next state by answering the subquestion. We incorporate the confidence of the state (i.e., answers in this case) as a reward. Specifically, we draw multiple sample answers from the world model, and use the proportion of the most frequent answer as the confidence. Higher confidence indicates that the state prediction is more consistent with the world knowledge of LLMs (Hao et al., 2023b), which typically leads to a more reliable reasoning step.

**Self-evaluation by the LLM.** It's sometimes easier to recognize the errors in reasoning than avoid generating them in advance. Thus, it's beneficial to allow the LLM to criticize itself with the question "Is this reasoning step correct?", and use the next-word probability of the token "Yes" as a reward. The reward evaluates LLM's own estimation of the correctness of reasoning. Note that the specific problems for self-evaluation can be different depending on the tasks.

**Task-specific heuristics.** RAP also allows us to flexibly plug in other task-specific heuristics into the reward function. For example, in plan generation for Blocksworld, we compare the predicted current state of blocks with the goal to calculate a reward (Section 4.1). The reward encourages the plan of movements to actively pace towards the target.

### 3.3 Planning with Monte Carlo Tree Search

Once equipped with the world model (Section 3.1) and rewards (Section 3.2), LLMs can reason with any planning algorithms. We adopt Monte Carlo Tree Search (MCTS) (Kocsis and Szepesvári, 2006; Coulom, 2007), a powerful planning algorithm that strategically explores the space of reasoning trees and strikes a proper balance between exploration and exploitation to find high-reward reasoning traces efficiently.

Specifically, MCTS builds a reasoning tree iteratively, where each node represents a state, and each edge represents an action and the transition from the current state to the next state after applying the action (Figure 1). To guide the LLM agent

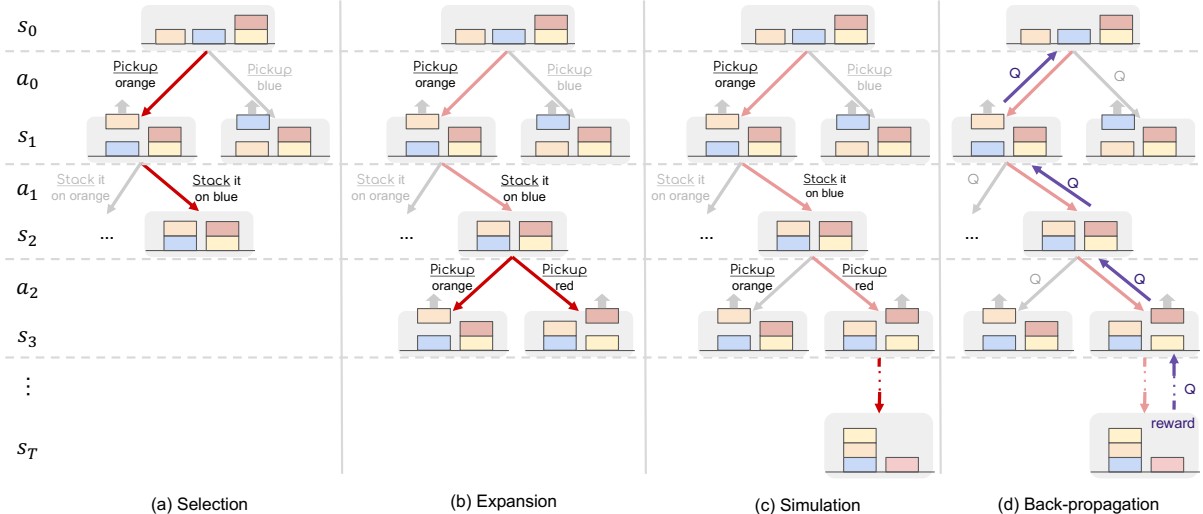

Figure 3: An illustration of the four phases in an iteration in MCTS planning (Section 3.3).

to expand and explore the most promising nodes of the tree, the algorithm maintains a state-action value function $Q : \mathcal{S} \times \mathcal{A} \mapsto \mathbb{R}$, where $Q(s, a)$ estimates the *expected future reward* of taking action $a$ in state $s$. Figure 3 illustrates four operations in each iteration to expand the tree and update $Q$ values. The process continues until a specified computational budget (e.g., the number of iterations) is reached, and the resulting traces are acquired from the tree. More details and the pseudo-code of the planning algorithm are given in Appendix A and Algorithm 1.

**Selection.** The first phase selects a portion of the existing tree that is most promising for further expansion in the next phase. Starting from the root node (i.e., initial state $s_0$), at each level of the tree, the algorithm selects a child node as the next node. The phase finishes when a leaf node of the current tree is reached. Figure 3(a) highlights the selected path in red. To balance between exploration (of less-visited nodes) and exploitation (of high-value nodes), we use the well-known *Upper Confidence bounds applied to Trees (UCT)* (Kocsis and Szepesvári, 2006) to select each child node. Specifically, at node $s$, we select the action in the tree by considering both the $Q$ value (for exploitation) and uncertainty (for exploration):

$$a^* = \arg \max_{a \in A(s)} \left[ Q(s, a) + w \sqrt{\frac{\ln N(s)}{N(c(s, a))}} \right], \quad (1)$$

where $N(s)$ is the number of times node $s$ has been visited in previous iterations, and $c(s, a)$ is the child node of applying $a$ in state $s$. The less a child node was visited before (i.e., the more uncertain about this child node), the higher the second term

in the equation. The weight $w$ controls the balance between exploration and exploitation.

**Expansion.** This phase expands the tree by adding new child nodes to the leaf node selected above. Given the state of the leaf node, we use the LLM (as agent) to sample $d$ possible actions (e.g., subquestions in math reasoning), and then use the LLM (as world model) to predict the respective next state, resulting in $d$ child nodes. Note that if the leaf node selected above is a terminal node (the end of a reasoning chain) already, we will skip expansion and jump to back-propagation.

**Simulation.** To estimate the expected future rewards ($Q$ values), this phase simulates the future situations of the current node using the world model. Starting from the current node as above, at each node $s$, we create an action following a *roll-out policy* and use the world model to predict the next state. The roll-out process continues until a terminal state is reached. There could be many ways to define the roll-out policy (e.g., by adding different randomness). In our experiments, for simplicity and reduced noises, we follow a similar process as in the expansion above, i.e., generating $d$ candidate actions and picking one of the largest local reward $a' = \max_{a'} r(s, a)$. In practice, for efficiency, we discard the computationally costly components in $r$ (e.g., the reward from the confidence of state requires sampling the answer multiple times), and use the resulting lightweight reward function for selecting actions during simulation.

**Back-propagation.** Once we reach a terminal state in the above phases, we obtain a reasoning path from the root node to the terminal node. We now back-propagate the rewards on the path to update

the $Q$ value of each state-action pair along the path. Specifically, we update $Q(s, a)$ by aggregating the rewards in all future steps of node $s$.

Once a predetermined number of MCTS iterations is reached, we terminate the algorithm and select the final reasoning trace from the constructed tree for evaluation. There are various ways for the selection. One is to start from the root node and iteratively choose the action with the highest $Q$ value until reaching a terminal. Also, one can directly select the path from the iterations that yielded the highest reward, or opt to choose the leaf node (and the respective root-to-leaf path) that has been visited the most. In practice, we observed that the second strategy often yields the best results.

## 3.4 RAP-Aggregation

For problems, such as math reasoning (Section 4.2) where only the final answer is required, RAP could produce multiple traces and answers from different MCTS iterations, which will be aggregated to produce the final answer. We refer to such a mechanism as RAP-Aggregation. Note that problems like plan generation or logical inference require a complete reasoning trace as output; thus, RAP-Aggregation will not be applied.

## 4 Experiments

In this section, we demonstrate the flexibility and effectiveness of our RAP framework by applying it to a wide range of problems, including plan generation in an embodied environment (4.1), mathematical reasoning for solving math word problems (4.2), and logical reasoning for verifying hypotheses (4.3). The subsequent sections demonstrate how the world model formulation in RAP enables a versatile design of the state and action, catering to various reasoning contexts.

We primarily compare RAP with chain-of-thought (CoT) (Wei et al., 2022), and its variants like least-to-most prompting (Zhou et al., 2022) as baselines. We also consider ensembling multiple reasoning paths if applicable (also known as self-consistency (Wang et al., 2022)). Moreover, we compare RAP with GPT-4 (OpenAI, 2023) when computation resources allow. By default, we use the LLaMA-33B model (Touvron et al., 2023a) as the base LLM for both our methods and baselines, with a sampling temperature of 0.8. All prompts are listed in Appendix C.

## 4.1 Plan Generation

The plan generation task aims to produce a sequence of actions to achieve a given goal, possibly with additional constraints. The ability to generate plans is important for intelligent embodied agents, e.g. household robots (Puig et al., 2018).

**Task setup.** To explore the viability of the RAP framework for plan generation tasks, we adapt and evaluate RAP on the Blocksworld benchmark (Valmeekam et al., 2022), where an agent is asked to rearrange the blocks into stacks in a particular order. We define a **state** as the current orientation of the blocks and an **action** as an instruction that moves blocks. Specifically, an action is composed of one of the 4 verbs (i.e., STACK, UNSTACK, PUT, and PICKUP) and manipulated objects. For the action space, we generate the currently valid actions given the domain restrictions on actions and the current orientation of the blocks. To transit between states, we take the current action and query the LLM to predict the state changes to the relevant blocks. We then update the current state by adding the new block conditions and removing the conditions that are no longer true. Once a state has met all conditions in the goal or the depth limit of the tree is reached, we terminate the associated node.

To assess the quality of actions within this domain, we use two separate **rewards**. First, we prompt the LLM with some example test cases along with their solutions, and then calculate the log probability of the action given the current state (*"Likelihood of action"* reward in Section 3.2), denoted as $r_1$. This reward reflects the intuition of the LLM as the reasoning agent. It's typically indicative when there are few steps left to the goal, while not as reliable for a distant goal. Additionally, we compare the new state after performing an action with the goal and provide a reward, $r_2$, scaling with the number of conditions met (*"Task-specific heuristics"* reward). Specifically, when all the conditions are met, we assign a super large reward to make sure this plan will be selected as the solution.

**Results.** We use test cases from the Blocksworld dataset (Valmeekam et al., 2023) and group them by minimum number of actions required, resulting in 30 cases solvable within 2 steps, 57 cases within 4 steps, and 114 cases within 6 steps. There are at most 5 blocks in each test case. As the baseline method, we prompt the LLM with 4 test cases with corresponding solutions, and ask it to generate a

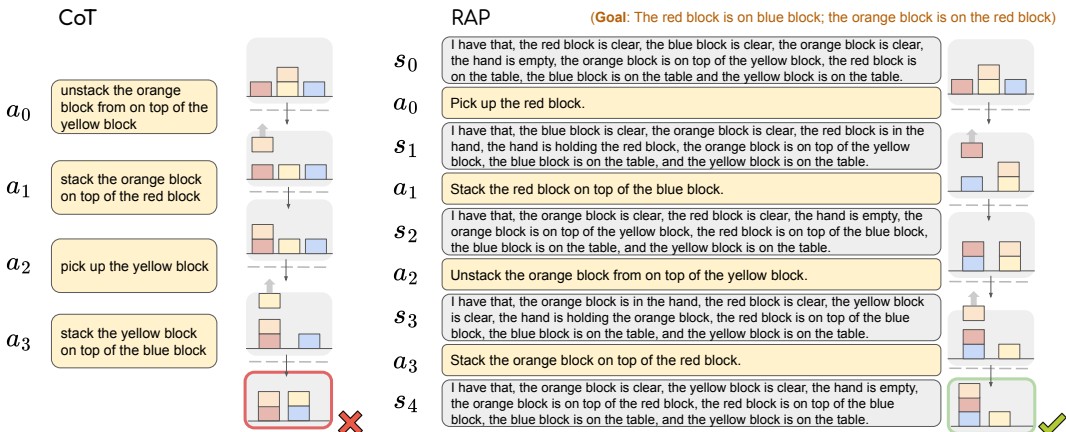

Figure 4: Comparing reasoning traces in Blocksworld from CoT (left) and RAP (right).

| Method | 2-step | 4-step | 6-step |
|---|---|---|---|
| CoT | 0.17 | 0.02 | 0.00 |
| CoT - pass@10 | 0.23 | 0.07 | 0.00 |
| CoT (GPT-4) | 0.50 | 0.63 | 0.40 |
| RAP$^{(10)}$ | 1.00 | 0.86 | 0.26 |
| RAP$^{(20)}$ | **1.00** | **0.88** | **0.42** |

Table 1: Results on Blocksworld. RAP$^{(10)}$ and RAP$^{(20)}$ refer to our method where the iteration number is set to 10 and 20, respectively. "pass@10" means 10 plans are sampled for each test case, and the test case is regarded as solved if at least one plan is correct. All other settings including RAP, only evaluate a single plan.

plan for a new question. This setting is the same as one described in Valmeekam et al. (2022), and we denote it as Chain-of-Thought (CoT) as the solution is generated step by step. For RAP, the same prompt is shown to help LLMs calculate $r_1$.

As shown in Table 1, CoT with LLaMA-33B can only generate successful plans for a few 2-step cases, and completely fails on harder problems. RAP substantially improves over CoT by nearly solving all problems within 4 steps, and a part of 6-step problems, achieving an average success rate of 64%. It's worth noting that the searching space of 6-step problems can be as large as $5^6$, while our algorithm can find a successful plan 42% of the time within 20 iterations. Even more, our framework allows LLaMA-33B to outperform GPT-4 by 33% relative gain, which is known to have much stronger reasoning ability (Bubeck et al., 2023).

**Case study.** We compare the reasoning paths from CoT and RAP in Figure 4. We summarize the reasons accounting for the improvement: (1) By maintaining the world state during reasoning, RAP can recognize valid actions for the current state, avoiding generating illegal plans. (2) RAP is capable of backtracking and trying out other solutions

when the first intuition from the LLM doesn't work. Specifically, CoT attempts to achieve the second goal, i.e. "orange on red", and achieve that with the first two steps. However, accomplishing the second goal first would prevent the first goal from being satisfied. On the contrary, even though RAP makes the same mistakes in the first iterations, our framework drives the agent to explore other possible paths (as described in Section 3.3) and finally generate a successful plan. (3) When calculating $r_t$, we can only feed the current state to the LLM and hide the history. E.g., in the case of Figure 4, to calculate the reward for $a_2$, the LLM is provided with a "new" test case, in which $s_2$ is the initial state. This significantly lowers the difficulties of the last few steps, and saves more iterations for harder decisions of the first few steps.

## 4.2 Math Reasoning

**Task setup.** Math reasoning tasks, such as GSM8k (Cobbe et al., 2021), often include a description and a final question. To arrive at the answer to the final question, it is necessary to undertake multi-step mathematical calculations based on the problem's context. It is thus natural to decompose the final question into a sequence of smaller sub-questions (Figure 2, right). We define a **state** as the values of intermediate variables, and an **action** as to propose an incremental sub-question about a unknown intermediate variable. The world model then responds to the sub-question using the intermediate variables and the problem description, adding the new intermediate variable value into the next state. We combine the self-evaluation of helpfulness by LLM $r_{t,1}$ and the confidence of state $r_{t,2}$ using weighted geometric mean $r_t = r_{t,1}^\alpha * r_{t,2}^{1-\alpha}$ as the **reward** function. This reward encourages more relevant and useful sub-questions. To account for the

| Method | Accuracy (%) |
|---|---|
| Chain-of-Thought | 29.4 |
| + SC$^{(10)}$ | 46.8 |
| Least-to-Most | 25.5 |
| + SC$^{(10)}$ | 42.5 |
| RAP$^{(1)}$ | 40.0 |
| RAP$^{(10)}$ | 48.6 |
| + aggr | **51.6** |

Table 2: Results on GSM8k. The superscripts indicate the number of samples or iterations.

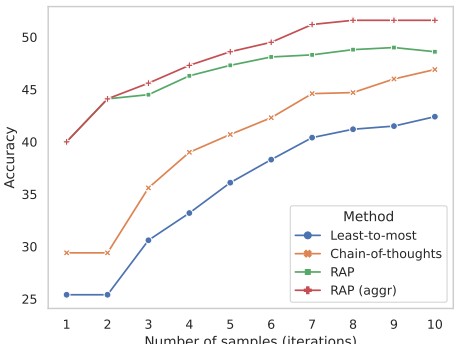

Figure 5: Results on GSM-8K, with different numbers of sampled paths or iterations.

impact of the reasoning path's length on the reward, we compute **the $Q$ value** by using the maximum of average rewards in future steps.

$$Q^*(s_t, a_t) = \max_{s_t, a_t, r_t, \ldots, s_l, a_l, r_l, s_{l+1}} \text{avg}(r_t, \ldots, r_l). \quad (2)$$

As a related work, Least-to-Most prompting (Zhou et al., 2022) shares a similar idea to us in sub-question decomposition, but they generate sub-questions all at once. On the contrary, RAP considers each action $a_t$ based on the current state $s_t$, which enables more informed decisions.

**Results.** We evaluate our framework on GSM8k, a dataset of grade school math word problems. We also evaluate the base model with CoT prompting (Wei et al., 2022), Least-to-Most prompting (Zhou et al., 2022), and their self-consistency (Wang et al., 2022) variants, as the baselines. We use the same 4-shot examples demonstrations for all methods.

As shown in Table 2, our RAP framework answers $48.8\%$ of the problems correctly, outperforming both the Chain-of-Thought and the Least-to-Most prompting with Self-Consistency. Notably, this result is achieved when RAP only selects only one reasoning trace based on the reward. The introduction of RAP-Aggregate further improves the accuracy by $\sim 3\%$. We also calculate the accuracy with different numbers of iterations in MCTS and self-consistency samples in baselines, as illustrated

| Method | Pred Acc | Proof Acc |
|---|---|---|
| CoT | 87.8 | 64.8 |
| CoT + SC | 89.8 | - |
| RAP (Ours) | **94.2** | **78.8** |

Table 3: Results on ProntoQA.

in Figure 5. We find that across all numbers of iterations/samples, RAP-Aggregation outperforms baselines consistently, which indicates that when only a few iterations/samples are allowed, our framework is significantly better at finding reliable reasoning paths with the guide of reward.

### 4.3 Logical Reasoning

**Task setup.** A logical reasoning task (e.g. PrOntoQA (Saparov and He, 2022)) typically provides a set of *facts* and *logical rules*, and a model is required to verify if a *hypothesis fact* is true or false by applying the logical rules to the given facts, as illustrated in Figure 2. These tasks not only require the correct final answer (true/false), but also a detailed proof demonstrating the result. To apply our framework, we define the **state** as a fact we are focusing on, analogous to the human's working memory (Baddeley, 1992) for inference. An **action** is defined as selecting a rule from the fact set. The world model performs a one-hop reasoning step to get a new fact as the next state. The **reward** is calculated with Self-evaluation (Section 3.2. Specifically, we prompt the LLM with a few examples with their labels to help it better understand the quality of reasoning steps. We use the average reward of future steps to update **the $Q$ function**, the same as Equation (2) for GSM8k.

**Results.** We assess the performance of our RAP framework on PrOntoQA (Saparov and He, 2022) and adopt their settings of "true" ontology (using real-world knowledge), "random" ordering of rules. We mix the examples requiring 3, 4, and 5 reasoning hops in a correct proof to prevent LLM from memorizing when to finish the reasoning. We sample 500 examples from the generation script released by Saparov and He (2022). We compare both the prediction accuracy of the final answer and the accuracy of the entire proof. We do 20 iterations for MCTS and 20 samples for self-consistency.

As the results presented in Table 3, our framework achieves a correct answer rate of 94.2% and a proof accuracy of 78.8%, surpassing the CoT baseline by 14% proof accuracy and the self-consistency CoT baseline by $4.4\%$ prediction accuracy. Such substantial improvements clearly

| Setting | Method | 2-step | 4-step | 6-step | 8-step | 10-step | 12-step | All |
|---------|--------|--------|--------|--------|--------|---------|---------|-----|
| Easy | CoT | 0.49 | 0.18 | 0.06 | 0.01 | 0.01 | 0.00 | 0.08 |
| | RAP$^{(10)}$ | 1.00 | 0.99 | 0.75 | 0.61 | 0.32 | 0.32 | 0.65 |
| Hard | CoT | 0.22 | 0.14 | 0.02 | 0.02 | 0.00 | 0.00 | 0.05 |
| | RAP$^{(10)}$ | 0.67 | 0.76 | 0.74 | 0.48 | 0.17 | 0.09 | 0.51 |

Table 4: Results on the full Blocksworld with Llama-2 70B.

demonstrate the effectiveness of RAP in solving logical reasoning problems in PrOntoQA. Also, as the case illustrated in Figure 2, RAP can effectively recognize when a reasoning chain comes to a dead end, and propagate the signal back to earlier reasoning steps, with the planning algorithm allowing it to explore alternatives to the previous steps. The self-evaluation reward further helps RAP to recognize potential incorrect reasoning steps, encouraging the agent to avoid them in future iterations.

## 5  Analysis

### 5.1  Complex problems

To further study whether RAP can help stronger LLMs to solve more complex problems, we conduct experiments on the full Blocksworld (Valmeekam et al., 2023) dataset using a more capable LLM, Llama-2 70B (Touvron et al., 2023b).

The full Blocksworld (Valmeekam et al., 2023) comprises 602 test cases. We group them based on the minimum number of actions required for each test case. Our experiments are conducted in two distinct settings: Easy and Hard. In Easy setting, we assume prior knowledge of the minimum number of actions for each case. Leveraging this information, we use demonstration cases that share the same minimum number of actions as the test case. For each group of cases, we randomly select 10 cases to create a pool of demonstration cases, leaving the remaining cases as the test set. During inference, we randomly sample 4-shot demonstration cases from this pool and utilize them to formulate prompts. In the Hard setting, we randomly select 10 cases from the full dataset to form a demonstration pool and subsequently exclude these cases from the test set. During inference, we randomly sample 4-shot demonstration cases from this global pool, irrespective of the minimum number of actions required for the test case.

We employ chain-of-thought prompting (Wei et al., 2022) as a baseline, and evaluate our RAP$^{(10)}$ (with 10 iterations) with an improved prompting technique (Appendix E). Our experimental results are summarized in Table 4. In both the Easy and

Hard settings, RAP demonstrates superior performance over CoT by a substantial margin. Notably, when the test case necessitates a larger number of steps (six or more) to solve, CoT exhibits a severe drop in success rates, whereas RAP maintains a relatively high success rate. Comparing these results with Section 4.1, we additionally conclude that RAP is a general framework able to enhance the reasoning abilities of LLMs, regardless of their intrinsic capabilities.

### 5.2  Reward Choice

In our main experiments, we choose the combination of rewards in our current experiments based on heuristics and our exploratory experiments. To understand the effects of the reward choice for LLM reasoning, we supplement comprehensive experiments on rewards for plan generation (Table 5) and math reasoning (Table 6).

Generally, the combination of multiple rewards contributes to the performance. However, the effects of a reward depends on the nature of tasks. For example, the action likelihood reward is essential for plan generation, but not very helpful to mathmatical reasoning. More discussions are in Appendix F.

## 6  Conclusion

In this paper, we present Reasoning via Planning (RAP), a novel LLM reasoning framework that equips LLMs with an ability to reason akin to human-like strategic planning. Our framework, which repurposes the LLM to act as both a world model and a reasoning agent, enables the LLM to simulate states of the world and anticipate action outcomes, and achieve an effective balance between exploration and exploitation via Monte-Carlo Tree Search. Extensive experiments on a variety of challenging reasoning problems demonstrate RAP's superiority over several contemporary CoT-based reasoning approaches, and even the advanced GPT-4 in certain settings.

## Limitations

In this work, we mainly focus on utilizing frozen LLMs, whose abilities might be bounded by the pre-training. In the future, it is worth exploring how to fine-tune LLMs to better reason and serve as a world model (Xiang et al., 2023), as well as how to combine external tools (Hao et al., 2023a; Schick et al., 2023) with RAP to solve more complex real-world problems.

## Ethics Statement

In this paper, we primarily focus on the applications on plan generation, mathematical reasoning, and logical reasoning, posing no significant ethical concerns. We recognize that future research on border applications of reasoning with LLMs may pose a risk of misuse, and we recommend careful consideration of all aspects of safety before relevant techniques are applied to the real world.

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

## A MCTS Planning

We adapt MCTS to search for the optimal reasoning path (Algorithm 1). Compared with traditional applications of MCTS, we are faced with a large reasoning space, and the heavy computational cost of LLMs. Thus, we made several modifications to the classic MCTS in our implementation: (1) For open domain problems, e.g., math problems, it's impossible to enumerate all actions (subquestions), so we reduce the action space by sampling a fixed number of potential actions from LLMs, conditioned on a prompt of the current state and in-context demonstration. (2) In the selection phase, if there are actions that haven't been visited before, we estimate the Q value with lightweight local rewards, e.g., self-evaluation reward, and then select the action with UCT. This provides prior knowledge for the exploration, which is crucial given the limited iteration budgets.

## B Experiment Settings

### B.1 Language Model Decoding

We use random sampling with a temperature of 0.8. The generation is cut off at the maximum length of 2048 or a newline token.

### B.2 Computing Resources

All of our experiments run on $4 \times$ NVIDIA A5000 GPUs with 24GB memory.

## C Prompt

### C.1 Plan Generation

We show the prompt to calculate the action likelihood for RAP below. The same prompt is also applied in CoT baseline. `<init_state>` and `<goals>` would be instantiated by the problem to solve.

```
I am playing with a set of blocks where
    I need to arrange the blocks into
    stacks. Here are the actions I can
    do

Pick up a block
Unstack a block from on top of another
    block
Put down a block
Stack a block on top of another block

I have the following restrictions on my
    actions:
```

```
I can only pick up or unstack one block
    at a time.
I can only pick up or unstack a block
    if my hand is empty.
I can only pick up a block if the block
    is on the table and the block is
    clear. A block is clear if the
    block has no other blocks on top of
    it and if the block is not picked
    up.
I can only unstack a block from on top
    of another block if the block I am
    unstacking was really on top of the
    other block.
I can only unstack a block from on top
    of another block if the block I am
    unstacking is clear.
Once I pick up or unstack a block, I am
    holding the block.
I can only put down a block that I am
    holding.
I can only stack a block on top of
    another block if I am holding the
    block being stacked.
I can only stack a block on top of
    another block if the block onto
    which I am stacking the block is
    clear.
Once I put down or stack a block, my
    hand becomes empty.

[STATEMENT]
As initial conditions I have that, the
    red block is clear, the yellow
    block is clear, the hand is empty,
    the red block is on top of the blue
    block, the yellow block is on top
    of the orange block, the blue block
    is on the table and the orange
    block is on the table.
My goal is to have that the orange
    block is on top of the red block.

My plan is as follows:

[PLAN]
unstack the yellow block from on top of
    the orange block
put down the yellow block
pick up the orange block
```

**Algorithm 1** RAP-MCTS

**Require:** Initial state $s_0$, state transition probability function $p_\theta$, reward function $r_\theta$, action generator $p_\phi$, number of generated actions $d$, depth limit $L$, number of roll-outs $N$, and exploration weight $w$

1: Initialize memory of actions $A : \mathcal{S} \mapsto \mathcal{A}$, children $c : \mathcal{S} \times \mathcal{A} \mapsto \mathcal{S}$ and rewards $r : \mathcal{S} \times \mathcal{A} \mapsto \mathbb{R}$
2: Initialize the state-action value function $Q : \mathcal{S} \times \mathcal{A} \mapsto \mathbb{R}$ and visit counter $N : \mathcal{S} \mapsto \mathbb{N}$
3: **for** $n \leftarrow 0, \dots, N - 1$ **do**
4:      $t \leftarrow 0$
5:      **while** $N(s_t) > 0$ **do**                       ▷ Selection
6:          $N(s_t) \leftarrow N(s_t) + 1$
7:          $a_t \leftarrow \arg\max_{a \in A(s_t)} \left[ Q(s_t, a) + w\sqrt{\frac{\ln N(s_t)}{N(c(s_t, a))}} \right]$
8:          $r_t = r(s_t, a_t), s_{t+1} \leftarrow c(s_t, a_t)$
9:          $t \leftarrow t + 1$
10:      **end while**
11:      **while** $s_t$ is not a terminal state $\wedge\ t \leq L$ **do**
12:          **for** $i \leftarrow 1, \dots, d$ **do**                 ▷ Expansion
13:              Sample $a_t^{(i)} \sim p_\phi(a \mid s_t)$, $s_{t+1}^{(i)} \sim p_\theta(s_t, a_t^{(i)})$, and $r_t^{(i)} \sim r_\theta(s_t, a_t^{(i)})$
14:              Update $A(s_t) \leftarrow \{a_t^{(i)}\}_{i=1}^d$, $c(s_t, a_t^{(i)}) \leftarrow s_{t+1}^{(i)}$, and $r(s_t, a_t) \leftarrow r_t^{(i)}$
15:          **end for**
16:          $a_{t+1} \leftarrow \arg\max_{a \in A(s_t)} r(s_t, a_t)$            ▷ Simulation
17:          $r_t \leftarrow r(s_t, a_t), s_{t+1} \leftarrow c(s_t, a_t)$
18:          $t \leftarrow t + 1$
19:      **end while**
20:      **for** $t' \leftarrow t, \dots, 0$ **do**                  ▷ Back propagation
21:          Update $Q(s_{t'}, a_{t'})$ with $\{r_{t'}, r_{t'+1}, \dots, r_t\}$
22:      **end for**
23: **end for**

---

stack the orange block on top of the
   red block
[PLAN END]

[STATEMENT]
As initial conditions I have that, the
   orange block is clear, the yellow
   block is clear, the hand is empty,
   the blue block is on top of the red
   block, the orange block is on top
   of the blue block, the red block is
   on the table and the yellow block
   is on the table.
My goal is to have that the blue block
   is on top of the red block and the
   yellow block is on top of the
   orange block.

My plan is as follows:

[PLAN]
pick up the yellow block
stack the yellow block on top of the
   orange block
[PLAN END]

[STATEMENT]
As initial conditions I have that, the
   red block is clear, the blue block

is clear, the orange block is
   clear, the hand is empty, the blue
   block is on top of the yellow
   block, the red block is on the
   table, the orange block is on the
   table and the yellow block is on
   the table.
My goal is to have that the blue block
   is on top of the orange block and
   the yellow block is on top of the
   red block.

My plan is as follows:

[PLAN]
unstack the blue block from on top of
   the yellow block
stack the blue block on top of the
   orange block
pick up the yellow block
stack the yellow block on top of the
   red block
[PLAN END]

[STATEMENT]
As initial conditions I have that, the
   red block is clear, the blue block
   is clear, the yellow block is
   clear, the hand is empty, the

yellow block is on top of the
orange block, the red block is on
the table, the blue block is on the
table and the orange block is on
the table.
My goal is to have that the orange
block is on top of the blue block
and the yellow block is on top of
the red block.

My plan is as follows:

[PLAN]
unstack the yellow block from on top of
the orange block
stack the yellow block on top of the
red block
pick up the orange block
stack the orange block on top of the
blue block
[PLAN END]

[STATEMENT]
As initial conditions I have that,
<initial_state>
My goal is to have that <goals>.

My plan is as follows:

[PLAN]

For the next state prediction with the world
model, we apply the prompts conditioned on the
last action. Here we show the prompt to update
the state after a "pick up" action as an example.
Again, <state> and <action> would be instanti-
ated with the current state and action.

I am playing with a set of blocks where
I need to arrange the blocks into
stacks. Here are the actions I can
do

Pick up a block
Unstack a block from on top of another
block
Put down a block
Stack a block on top of another block

I have the following restrictions on my
actions:
I can only pick up or unstack one block
at a time.

I can only pick up or unstack a block
if my hand is empty.
I can only pick up a block if the block
is on the table and the block is
clear. A block is clear if the
block has no other blocks on top of
it and if the block is not picked
up.
I can only unstack a block from on top
of another block if the block I am
unstacking was really on top of the
other block.
I can only unstack a block from on top
of another block if the block I am
unstacking is clear. Once I pick up
or unstack a block, I am holding
the block.
I can only put down a block that I am
holding.
I can only stack a block on top of
another block if I am holding the
block being stacked.
I can only stack a block on top of
another block if the block onto
which I am stacking the block is
clear. Once I put down or stack a
block, my hand becomes empty.

After being given an initial state and
an action, give the new state after
performing the action.

[SCENARIO 1]
[STATE 0] I have that, the white block
is clear, the cyan block is clear,
the brown block is clear, the hand
is empty, the white block is on top
of the purple block, the purple
block is on the table, the cyan
block is on the table and the brown
block is on the table.
[ACTION] Pick up the brown block.
[CHANGE] The hand was empty and is now
holding the brown block, the brown
block was on the table and is now
in the hand, and the brown block is
no longer clear.
[STATE 1] I have that, the white block
is clear, the cyan block is clear,
the brown block is in the hand, the
hand is holding the brown block,

the white block is on top of the
purple block, the purple block is
on the table and the cyan block is
on the table.

[SCENARIO 2]
[STATE 0] I have that, the purple block
is clear, the cyan block is clear,
the white block is clear, the hand
is empty, the white block is on top
of the brown block, the purple
block is on the table, the cyan
block is on the table and the brown
block is on the table.
[ACTION] Pick up the cyan block.
[CHANGE] The hand was empty and is now
holding the cyan block, the cyan
block was on the table and is now
in the hand, and the cyan block is
no longer clear.
[STATE 1] I have that, the cyan block
is in the hand, the white block is
clear, the purple block is clear,
the hand is holding the cyan block,
the white block is on top of the
brown block, the purple block is on
the table and the brown block is on
the table.

[SCENARIO 3]
[STATE 0] <state>
[ACTION] <action>
[CHANGE]

### C.2 Math Reasoning

We show the prompt of RAP for math reasoning
as below. The prompt is used for both action pro-
posal and next state prediction. After instantiate
<question>, we append a prefix Question 5.1 to
the prompt, so that we can sample the first action
with the LLM. The future actions are sampled sim-
ilarly, except that all previous sub-questions and
sub-answers need to be appended to the prompt,
following the formats of in-context demonstration.
The next state prediction, i.e., answering the sub-
question, works in the same way.

Given a question, please decompose it
into sub-questions. For each
sub-question, please answer it in a
complete sentence, ending with "The
answer is". When the original

question is answerable, please
start the subquestion with "Now we
can answer the question: ".

Question 1: Four years ago, Kody was
only half as old as Mohamed. If
Mohamed is currently twice as 30
years old, how old is Kody?
Question 1.1: How old is Mohamed?
Answer 1.1: He is currently 30 * 2 = 60
years old. The answer is 60.
Question 1.2: How old was Mohamed four
years ago?
Answer 1.2: Four years ago, he must
have been 60 - 4 = 56 years old.
The answer is 56.
Question 1.3: How old was Kody four
years ago?
Answer 1.3: Kody was half as old as
Mohamed four years ago. Thus, Kody
was 56 / 2 = 28 years old. The
answer is 28.
Question 1.4: Now we can answer the
question: How old is Kody?
Answer 1.4: She is currently 28 + 4 =
32 years old. The answer is 32.

Question 2: On a moonless night, three
fireflies danced in the evening
breeze. They were joined by four
less than a dozen more fireflies
before two of the fireflies flew
away. How many fireflies remained?
Question 2.1: How many fireflies joined?
Answer 2.1: The fireflies were joined
by four less than a dozen more
fireflies, which are 12 - 4 = 8
fireflies. The answer is 8.
Question 2.2: Now we can answer the
question: How many fireflies
remained?
Answer 2.2: Three fireflies were
dancing originally. They were
joined by 8 fireflies before two of
them flew away. So there were 3 + 8
- 2 = 9 remaining. The answer is 9.

Question 3: Ali has four $10 bills and
six $20 bills that he saved after
working for Mr. James on his farm.
Ali gives her sister half of the

total money he has and uses 3/5 of
the remaining amount of money to
buy dinner. Calculate the amount of
money he has after buying the
dinner.
Question 3.1: How much money does Ali
have in total?
Answer 3.1: Ali has four $10 bills and
six $20 bills. So he has 4 * 10 + 6
* 20 = 160 dollars. The answer is
160.
Question 3.2: How much money does Ali
give to his sister?
Answer 3.2: Ali gives half of the total
money he has to his sister. So he
gives 160 / 2 = 80 dollars to his
sister. The answer is 80.
Question 3.3: How much money does Ali
have after giving his sister the
money?
Answer 3.3: After giving his sister the
money, Ali has 160 - 80 = 80
dollars left. The answer is 80.
Question 3.4: How much money does Ali
use to buy dinner?
Answer 3.4: Ali uses 3/5 of the
remaining amount of money to buy
dinner. So he uses 80 * 3/5 = 48
dollars to buy dinner. The answer
is 48.
Question 3.5: Now we can answer the
question: How much money does Ali
have after buying the dinner?
Answer 3.5: After buying the dinner,
Ali has 80 - 48 = 32 dollars left.
The answer is 32.

Question 4: A car is driving through a
tunnel with many turns. After a
while, the car must travel through
a ring that requires a total of 4
right-hand turns. After the 1st
turn, it travels 5 meters. After
the 2nd turn, it travels 8 meters.
After the 3rd turn, it travels a
little further and at the 4th turn,
it immediately exits the tunnel. If
the car has driven a total of 23
meters around the ring, how far did
it have to travel after the 3rd
turn?

Question 4.1: How far did the car
travel except for the 3rd turn?
Answer 4.1: It travels 5 meters after
the 1st, 8 meters after the 2nd,
and 0 meters after the 4th turn.
It's a total of 5 + 8 + 0 = 13
meters. The answer is 13.
Question 4.2: Now we can answer the
question: How far did the car have
to travel after the 3rd turn?
Answer 4.2: The car has driven a total
of 23 meters around the ring. It
travels 13 meters except for the
3rd turn. So it has to travel 23 -
13 = 10 meters after the 3rd turn.
The answer is 10.

Question 5: <question>

## C.3  Logical Reasoning

We show the prompt for action proposal, action
likelihood calculation, and next state prediction.
<fact> and <query> would be instantiated with
the problem.

Given a list of facts, and a current
claim, output one possible fact as
the next step. Be sure to copy the
exact sentences in the facts. Do
not change any wording. Do not
create your own words.

Facts 1: Each lepidopteran is an
insect. Each arthropod is a
protostome. Every animal is
multicellular. Protostomes are
invertebrates. Each whale is bony.
Each painted lady is a butterfly.
Invertebrates are animals.
Butterflies are lepidopterans. Each
insect is six-legged. Every insect
is an arthropod. Arthropods are not
bony.
Query 1: True or false: Sally is not
bony.
Claim 1.1: Sally is an insect.
Next 1.1: Each insect is six-legged.
Claim 1.2: Sally is a butterfly.
Next 1.2: Butterflies are lepidopterans.
Claim 1.3: Sally is a lepidopteran.
Next 1.3: Each lepidopteran is an
insect.

Claim 1.4: Sally is not bony.
Next 1.4: Finish.
Claim 1.5: Sally is an arthropod.
Next 1.5: Arthropods are not bony.
Claim 1.6: Sally is a painted lady.
Next 1.6: Each painted lady is a
    butterfly.

Facts 2: Prime numbers are natural
    numbers. Every Mersenne prime is
    not composite. Imaginary numbers
    are not real. Every real number is
    a number. Natural numbers are
    integers. Every real number is
    real. Every Mersenne prime is a
    prime number. Natural numbers are
    positive. Prime numbers are not
    composite. Integers are real
    numbers.
Query 2: True or false: 127 is not real.
Claim 2.1: 127 is real.
Next 2.1: Finish.
Claim 2.1: 127 is a natural number.
Next 2.1: Natural numbers are integers.
Claim 2.2: 127 is a prime number.
Next 2.2: Prime numbers are natural
    numbers.
Claim 2.3: 127 is a real number.
Next 2.3: Every real number is real.
Claim 2.4: 127 is a Mersenne prime.
Next 2.4: Every Mersenne prime is a
    prime number.
Claim 2.5: 127 is an integer.
Next 2.5: Integers are real numbers.

Facts 3: Lepidopterans are insects.
    Every animal is multicellular. Each
    insect is an arthropod. Each
    invertebrate is an animal. Insects
    are six-legged. Arthropods are
    small. Arthropods are
    invertebrates. Each butterfly is a
    lepidopteran. Whales are not small.
Query 3: True or false: Polly is not
    small.
Claim 3.1: Polly is an arthropod.
Next 3.1: Arthropods are small.
Claim 3.2: Polly is an insect.
Next 3.2: Each insect is an arthropod.
Claim 3.3: Polly is small.
Next 3.3: Finish.

Claim 3.4: Polly is a lepidopteran.
Next 3.4: Lepidopterans are insects.

Facts 4: Every cat is a feline. Mammals
    are vertebrates. Bilaterians are
    animals. Vertebrates are chordates.
    Carnivores are mammals. Mammals are
    not cold-blooded. Each chordate is
    a bilaterian. Every feline is a
    carnivore. Snakes are cold-blooded.
    Animals are not unicellular. Every
    carnivore is not herbivorous.
Query 4: True or false: Fae is not
    cold-blooded.
Claim 4.1: Fae is a feline.
Next 4.1: Every feline is a carnivore.
Claim 4.2: Fae is not cold-blooded.
Next 4.2: Finish.
Claim 4.2: Fae is a mammal.
Next 4.2: Mammals are not cold-blooded.
Claim 4.3: Fae is a cat.
Next 4.3: Every cat is a feline.
Claim 4.4: Fae is a carnivore.
Next 4.4: Carnivores are mammals.

Facts 5: Prime numbers are prime. Real
    numbers are numbers. Every integer
    is a real number. Real numbers are
    not imaginary. Mersenne primes are
    prime numbers. Complex numbers are
    imaginary. Each prime number is a
    natural number. Natural numbers are
    positive. Each Mersenne prime is
    prime. Each natural number is an
    integer.
Query 5: True or false: 7 is imaginary.
Claim 5.1: 7 is not imaginary.
Next 5.1: Finish.
Claim 5.1: 7 is a natural number.
Next 5.1: Each natural number is an
    integer.
Claim 5.2: 7 is a prime number.
Next 5.2: Each prime number is a
    natural number.
Claim 5.3: 7 is a real number.
Next 5.3: Real numbers are not
    imaginary.
Claim 5.4: 7 is an integer.
Next 5.4: Every integer is a real
    number.

```
Facts 6: Spiders are not six-legged.
   Insects are six-legged. Insects are
   arthropods. Every animal is not
   unicellular. Invertebrates are
   animals. Lepidopterans are insects.
   Every arthropod is segmented.
   Arthropods are invertebrates. Every
   butterfly is a lepidopteran. Stella
   is a butterfly.
Query 6: True or false: Stella is
   six-legged.
Claim 6.1: Stella is an insect.
Next 6.1: Insects are six-legged.
Claim 6.2: Stella is a lepidopteran.
Next 6.2: Lepidopterans are insects.
Claim 6.3: Stella is a butterfly.
Next 6.3: Every butterfly is a
   lepidopteran.
Claim 6.4: Stella is six-legged.
Next 6.4: Finish.

Facts 7: <fact>
Query 7: <query>
```

Table 5: Ablation study on Blocksworld. $R_1$ is action likelihood reward, $R_2$ is task-specific reward, and $R_3$ is self-evaluation reward.

| $R_1$ | $R_2$ | $R_3$ | Success |
|---|---|---|---|
| ✓ | ✓ | ✗ | 0.88 |
| ✓ | ✓ | ✓ | 0.91 |
| ✓ | ✗ | ✗ | 0.46 |
| ✗ | ✓ | ✗ | 0.21 |
| ✗ | ✗ | ✓ | 0.14 |
| ✗ | ✗ | ✗ | 0.02 |

Table 6: Ablation study on GSM8k (first 300 examples). $R_1$ is state transition confidence reward, $R_2$ is action likelihood reward, and $R_3$ is self-evaluation reward.

| $R_1$ | $R_2$ | $R_3$ | RAP$^{(1)}$ | RAP$^{(10)}$ | +aggr |
|---|---|---|---|---|---|
| ✓ | ✗ | ✓ | 0.410 | 0.450 | 0.503 |
| ✓ | ✗ | ✗ | 0.350 | 0.447 | 0.490 |
| ✓ | ✓ | ✗ | 0.373 | 0.423 | 0.443 |

## D  Related work: world model and planning

Recent years have witnessed successful applications of planning algorithms (Sekar et al., 2020), such as AlphaZero (Silver et al., 2017), and MuZero (Schrittwieser et al., 2020). These algorithms are typically based on tree-structured search and are designed to effectively maintain the balance of exploration and exploitation. Knowledge of transition dynamics is the prerequisite for planning, and recent research on model-based reinforcement learning propose to learn a world model (or dynamics model) to plan or assist policy learning. To improve sample efficiency, previous research attempts to learn a world model from offline trajectories, and directly learn a policy within the world model (Ha and Schmidhuber, 2018a,b). With latent imagination in a world model, RL agents can be trained to solve long-horizon tasks (Hafner et al., 2019, 2020). Besides, the world model is also shown to be helpful to physical robot learning (Wu et al., 2023). In this paper, we use LLMs as world models and apply a planning algorithm to search for a reasoning path. This is similar in spirit to model predictive control (Camacho and Alba, 2013). Compared with previous works, our framework uses general LLMs as the world model and can be adapted to a wide range of open-domain reasoning tasks. Xiang et al. (2023) propose to train LLMs wih a external world model to gain embodied experience, while RAP focuses on the inference stage and is compatible with any training methods.

## E  Adaptive Prompting

Through our preliminary experiments, we observed that the performance of LLMs is impacted by the discrepancy in difficulty between demonstration cases and the test cases. In the case of RAP, when a new state is predicted, we reformulate the remaining task as a new test case, initialized with the predicted new state. This new test case would require a smaller minimum number of actions, leading to a disparity in the distribution of the demonstration cases and the new cases. To mitigate this issue, we pre-compute the intermediate states of the demonstration cases beforehand. During inference, we truncate the trace from the beginning for each new state in an iteration, which reduces the minimum action number of the demonstration cases as the search tree deepens. This technique significantly enhances the performance of RAP, especially for more intricate problems, which are more susceptible to distribution mismatches.

## F  Reward Choice

**Results.** We conduct comprehensive experiments on rewards for plan generation (Table 5) and math reasoning (Table 6). Note that, in both tables, the first row indicates the setting we use in the main experiments. As shown in Table 5, the combination of action likelihood and task-specific reward (row 1) can significantly outperform the single reward baselines (row 3, 4, 5). Interestingly, adding the self-evaluation reward can further improve the performance slightly (row 2). Furthermore, as the results on the first 300 samples of GSM8k shown in Table 6, we can see adding either action likelihood (row 3) or self-evaluation (row 1) on top of confidence reward (row 2) can boost the RAP performance of only using confidence reward (row 1) with one iteration, but action likelihood reward downgrades the accuracy with more iterations. The self-evaluation reward leads to the best performance overall. This indicates the importance of self-evaluation reward in guiding reasoning as an effective and computationally efficient prior to exploration.

**Self-evaluation and action likelihood.** The rewards of self-evaluation and action likelihood are of particular interest, as they can be applied to a wide range of reasoning tasks. Generally, the best usage and combination with other rewards require empirical design and understanding of the task nature, and their effectiveness can vary significantly across different tasks. Here, we provide some intuitions behind the reward choices:

(a) For the problems in which one reasoning step is short and structured, the action likelihood can be very indicative. Otherwise, it may be disturbed by unimportant tokens and become unreliable. For instance, a single step within the Blocksworld domain typically adheres to specific patterns (e.g., PICK/PUT/STACK a block...), rendering the action likelihood indicative. However, in the math domain, a reasoning step is expressed in natural language sentences, allowing for greater freedom and potentially introducing noise.

(b) For the problems where it's easier to recognize some errors afterward than avoid them during generation, self-evaluation emerges as a helpful mechanism for enhancing reasoning accuracy. In mathematical reasoning, LLMs may struggle to generate a correct reasoning step in the first place, but the detection of calculation or logic errors is more feasible. In Blocksworlds, however, assessing the quality of a candidate action is not straightforward and still requires multi-step reasoning. This characteristic diminishes the accuracy of the self-evaluation reward, making it less helpful especially given that likelihood already provides a good intuition for search.