# OpenReview forum: "Reasoning with Language Model is Planning with World Model"
_EMNLP/2023/Conference — EMNLP 2023 Main_

### Official Review · Reviewer_rY4T · 2023-08-05

**Soundness:** 3

**Excitement:**

3: Ambivalent: It has merits (e.g., it reports state-of-the-art results, the idea is nice), but there are key weaknesses (e.g., it describes incremental work), and it can significantly benefit from another round of revision. However, I won't object to accepting it if my co-reviewers champion it.

**Paper Topic And Main Contributions:**

The paper proposed a method for the reasoning problems. It leverages large language models to propose an open-ended sets of action space and next state transaction to build a tree-structured internal world model, and uses the Monte Carlo Tree search to find the most optimal planning path. It incrementally builds the reasoning tree, and record rewards for each internal nodes through multiple rounds of simulation. The paper evaluates the method on three different reasoning tasks: Blocksworld, high school math, and logical reasoning, and demonstrated improved performance compared to baselines.


**Questions For The Authors:**

a. It would be helpful to include a full sample prompt to assist method description

b. It was unclear if the action and space space of each step is pre-defined through prompt or completely open-ended

c. Section 3.2 "Likelihood of the action": it is unclear if the likelihood is a number output by the LLM with prompts or through other methods

d. Ln 273: how did the paper draw "multiple sample answers" from the world model for each state?

e. Section 3.2 "Task-specific heuristics": how much of the method requires task-specific manual curation?

f. Ln 356: how to define "terminal state"

g. Table 1: for fair a comparison, if RAP was allowed with multiple trial runs, why aren't CoTs allowed with multiple trials with added small randomness?

h. Ln 455: "super large" -- how large?

i. Ln 524: what is alpha?

j. Ln 541: what is self-consistency?

k. Figure 5: by the trajectory, if the number of sample keeps increasing, could Least-to-most or CoT outperform RAP?



**Reasons To Accept:**

The paper proposed a new method for reasoning that leverages large language models and explicitly creating internal simulation trees for the optimal decision making. The method combines the open-ended LLM action and state prediction capabilities and deliberate Monte Carlo Tree search reasoning abilities to improve the reasoning performance of the models. The method introduced several reward designs, action selection, tree expansion and decision simulation & aggregation to enhance the performance. The paper conducted thorough experimentations of the method across three different tasks and several baselines of each, and demonstrated improved performance.


**Reasons To Reject:**

The method relies heavily on the outputs of the large language model, the performance of which is questionable. Both the action space and the next state prediction depends on the output of LLM, which the paper did not investigate how faithful these outputs are. More importantly, one of the rewards is a self-evaluation by the LLM, which could deviate a lot from the actual fact.

There are several important details about the methods (details below) that could be further explained.

The Limitations section could be strengthened.


**Reproducibility:**

3: Could reproduce the results with some difficulty. The settings of parameters are underspecified or subjectively determined; the training/evaluation data are not widely available.

**Reviewer Confidence:**

4: Quite sure. I tried to check the important points carefully. It's unlikely, though conceivable, that I missed something that should affect my ratings.

---

> ### Author Rebuttal · Authors · 2023-08-29
>
> We thank reviewer rY4T for acknowledging that our proposed method is new and our experiments are thorough for demonstrating the improved performance across different tasks and baselines. We address all the questions and concerns of the reviewer as follows:
>
> ## On the questionable performance of LLMs
>
> First, we kindly refer the reviewer to check our previous responses to the reviewer eX3j (the first paragraph) regarding why we still use LLMs as the world models given their imperfection. We’d like to emphasize that our framework is designed to **improve the reasoning ability of such (imperfect) LLMs**.
>
> Regarding the concern that the action space and next state generated by LLM may not guarantee faithfulness, we would like to point out that it’s infeasible to investigate the faithfulness of each action and state given the large problem set and sample space, but their effectiveness are **already demonstrated by the task performance improvement** in our experiments. Moreover, to robustly search for the most plausible chain of reasoning within the realm of possibilities offered by the LLMs, we further propose the use of Monte Carlo Tree Search (MCTS), which is **precisely aimed to prune the unfaithful action and states**, which indeed leads to significant improvement.
>
> Finally, we hope the mistrust of LLMs’ probabilistic outputs doesn't overshadow the appreciation of our true contributions in this paper, which is to leverage a new introduction of the world model to novelly reformulate the reasoning problem as a planning task. Rather than viewing our work as a technical addition like "combining LLM’s action and state prediction and MCTS to improve reasoning performance", we aspire to offer insights into autonomous agent design, bridging the intuition from cognitive science and techniques from reinforcement learning. By formally introducing this reformulation, as the first work in this field, we hope to open up a promising new avenue for future research, as also acknowledged by other reviewers.
>
> ## Self-Evaluation Reward
>
> Part of the concern from the reviewer about the self-evaluation reward is shared with the concerns we addressed above. However, there are a few additional points we’d like to bring up.
>
> First, using LLM’s own feedback to guide the LLM’s behavior as the self-evaluation has been heavily studied recently, see [2, 3, 4, 5] and even a survey paper [5] on this topic. Our design of self-evaluation reward is motivated by the intuition that it’s easier to recognize some errors afterward than avoid them during generation.
>
> Second, to further address these concerns, we have conducted two additional ablation experiments (please refer to our response to Reviewer p5Yn), which **underscore the effectiveness of self-evaluation reward**. In particular, Blocksworld experiences further performance improvement with self-evaluation (row2 over row1 in RTable 1), and is similar as of GSM8k, especially with limited iterations (row2 over row1 in RTable 2).
>
> Additionally, It’s worth clarifying that the rewards we describe are solely used to guide the search, but not used in the final correctness evaluation of the outcome of our method and the baselines.
>
> ## Explanation of details
>
> We thank the reviewer for the detailed questions and comments. We carefully address them all with additional explanations and experiment results as follows:
>
> a. The prompts are included in supplementary materials (line 897). We will also present some prompts in the appendix of the updated paper.
>
> b. The action space is open-ended, i.e., sampled from its generative distribution (line 218)
>
> c. We calculate the likelihood of generating an action by averaging the likelihood of each token measured by the LLM.
>
> d. We randomly decode multiple times (in contrast to greedy decoding) to get multiple predictions
>
> e. In our experiment, only Blocksworld makes use of the task-specific reward. We want to emphasize that RAP is a general framework that can be applied to various tasks. Since some reasoning tasks have their own natural reward heuristics (e.g. how many subgoals are achieved in Blocksworld), RAP is flexible enough to make use of any task-specific reward heuristics.
>
> f. A node that has gathered all information for a reasoning problem is a terminal node (e.g. when the predicted state is equivalent to the goal state in Blocksworld; when LLM answers the original question in GSM8k/PrOntoQA). Besides, a node reaching search depth limitation is also considered terminal.
>
> g. We include “CoT - pass@10” in Table 1, where CoT is allowed to sample 10 answers, and considered success if any of the samples is correct. Note that this is an **easier setting and not a fair comparison**, because RAP must submit only one single plan for evaluation.
>
> h. We set it to 100 in our experiments (which is of a much larger scale than normal reward), but the effect is the same as setting it to “infinity”, because it’s large enough for the agent to select the current reasoning chain.
>
> i. It’s a hyperparameter to control the weight of the geometric mean of $r_1$ and $r_2$. We use $\alpha=0.5$ in experiments.
>
> j. Self-consistency [1] is an improvement over CoT prompting. It’s sampling multiple reasoning chains and returns the answer based on a majority vote.
>
> k. We run an additional experiment and find Chain-of-Thoughts + SC$^{(15)}$ achieves the accuracy of 0.515, which is comparable to RAP$^{(10)}$+aggr. This indicates our framework can explore the reasoning space more efficiently with fewer iterations.
>
> We hope our additional experiment results and discussions/explanations/clarifications could further strengthen the soundness of our paper, and the appreciation of its core contributions could ignite excitement for the promising directions revealed by our work. We will include the discussions above, and extend the limitation section into our updated version. Thanks for your questions and suggestions.
>
> ## Reference
>
> [1] Wang, X., Wei, J., Schuurmans, D., Le, Q., Chi, E., Narang, S., Chowdhery, A. and Zhou, D., 2022. Self-consistency improves chain of thought reasoning in language models. arXiv preprint arXiv:2203.11171.
>
> [2] Welleck, Sean, Ximing Lu, Peter West, Faeze Brahman, Tianxiao Shen, Daniel Khashabi, and Yejin Choi. "Generating sequences by learning to self-correct." arXiv preprint arXiv:2211.00053 (2022).
>
> [3] Shinn, Noah, Beck Labash, and Ashwin Gopinath. "Reflexion: an autonomous agent with dynamic memory and self-reflection." arXiv preprint arXiv:2303.11366 (2023).
>
> [4] Paul, Debjit, Mete Ismayilzada, Maxime Peyrard, Beatriz Borges, Antoine Bosselut, Robert West, and Boi Faltings. "Refiner: Reasoning feedback on intermediate representations." arXiv preprint arXiv:2304.01904 (2023).
>
> [5] Xie, Yuxi, Kenji Kawaguchi, Yiran Zhao, Xu Zhao, Min-Yen Kan, Junxian He, and Qizhe Xie. "Decomposition enhances reasoning via self-evaluation guided decoding." arXiv preprint arXiv:2305.00633 (2023).
>
> [6] Pan, Liangming, Michael Saxon, Wenda Xu, Deepak Nathani, Xinyi Wang, and William Yang Wang. "Automatically Correcting Large Language Models: Surveying the landscape of diverse self-correction strategies." arXiv preprint arXiv:2308.03188 (2023).

---

### Official Review · Reviewer_eX3j · 2023-08-05

**Soundness:** 4

**Excitement:**

4: Strong: This paper deepens the understanding of some phenomenon or lowers the barriers to an existing research direction.

**Paper Topic And Main Contributions:**

The paper proposes the Reasoning via Planning (RAP) framework to enhance large language models' reasoning capabilities by incorporating an external world model and principled planning.

Main Contributions:
1. Introducing RAP Framework: RAP augments large language models with a world model and uses principled planning (MCTS) for reasoning.
2. Enhancing Reasoning: By simulating the state of the world and guiding the reasoning process, RAP improves performance in complex tasks compared to existing methods like CoT.
3. General Applicability: RAP is a versatile framework applicable to various challenging problems, consistently outperforming strong baselines in tasks like Blocksworld, math reasoning, and logical inference.

**Reasons To Accept:**

1. I think the idea presented in this paper is novel. Utilizing world models to enhance existing large language models is a promising direction, and this work takes a step forward in this direction.
2. The paper introduces several well-explored designs for rewards, enabling the utilization of traditional planning methods like MCTS, which is a valuable exploration.
3. Significant performance improvements across multiple tasks demonstrate the generality and effectiveness of the proposed approach.

**Reasons To Reject:**

1. Although LLMs demonstrate flexibility in modeling world models, their inherent limitations in logic and mathematics can lead to inaccuracies when used for simulating logic and mathematics.
2. The designs of several rewards are relatively intuitive, but they do not provide strong guarantees for the performance of the method.
3. As described by the authors in the Limitations section, this work uses frozen LLMs without fine-tuning, which is constrained by the capabilities of the LLMs itself.

**Reproducibility:**

4: Could mostly reproduce the results, but there may be some variation because of sample variance or minor variations in their interpretation of the protocol or method.

**Reviewer Confidence:**

4: Quite sure. I tried to check the important points carefully. It's unlikely, though conceivable, that I missed something that should affect my ratings.

---

> ### Author Rebuttal · Authors · 2023-08-29
>
> We thank the reviewer eX3j for acknowledging that our idea is novel, taking a step forward in a promising direction, our reward designs are well-explored as a valuable exploration to leverage traditional planning algorithms, and our performance improvement is significant across multiple tasks demonstrating the generality and effectiveness of the proposed approach. We address all the questions and concerns from the reviewer as follows:
>
> ## LLM’s inherent limitations in logic and mathematics
> We agree with the reviewer that current LLMs are imperfect, which is exactly the motivation behind our proposed novel framework, aiming to enhance the reasoning abilities of LLMs and to be generally applied to improve most LLMs (either black-box or open-sourced ones, also pointed out by the reviewer p5Yn). On the other hand, building a “perfect” world model generally for various reasoning tasks (not handcrafting task-specific environments) hasn’t been solved yet, which requires a significant amount of research effort potentially behind a whole new research direction. Therefore, repurposing a reasonably large LLM (LLaMA-33B) as the world model is not only a cost-efficient choice, but also enjoys the flexibility of modeling the world state using language, which is considered as the core contributions of our work. While such a world model isn’t flawless and requires extensive future work for further improvement, in this single paper, we clearly demonstrate its significant effectiveness via comprehensive experiments/tasks and robust results (as pointed out by all reviewers).
>
> Besides, when there are other ways to enhance LLMs’ reasoning abilities, like using calculators to improve their mathematical abilities [1], our proposed method is complementary to them and can directly benefit from such enhancements. We believe it’s also an interesting future work to integrate additional techniques to further enhance LLM reasoning.
>
> ## On reward designs
>
> We appreciate the reviewer’s comments on our reward designs. While the reward design in our paper requires task-specific heuristics and some exploratory experiments (similar to what reward engineering in RL does), our thorough experiments demonstrate that well-designed rewards can boost the reasoning abilities of LLMs across various tasks. Please refer to our response to Reviewer p5Yn to see two new comprehensive ablation strides on rewards, which further show the effectiveness of each reward design. We hope our new additional results can answer the reviewer’s question and demonstrate the effectiveness of our reward design empirically. We will supplement these analyses of reward designs in the updated paper.
>
> ## Using frozen LLMs without fine-tuning
> We thank the reviewer for carefully checking our limitation section. We’d like to point out the significance of developing methods utilizing frozen LLMs. This has actually been an important research topic, exemplified by Chain-of-thoughts and many other papers [2, 3]. Realistically, users often don't have the luxury of fine-tuning an LLM before application. As suggested by Reviewer p5Yn, our approach enjoys generality across most black box language models without necessitating the fine-tuning process. However, we acknowledge that leveraging the world model for further fine-tuning the LLM could potentially further enhance performance if we collect the data and design the task properly, which is out of the scope of this paper and seen as a promising future direction.
>
> ## Reference
>
> [1] Schick, Timo, Jane Dwivedi-Yu, Roberto Dessì, Roberta Raileanu, Maria Lomeli, Luke Zettlemoyer, Nicola Cancedda, and Thomas Scialom. "Toolformer: Language models can teach themselves to use tools." arXiv preprint arXiv:2302.04761 (2023).
>
> [2] Wei, Jason, Xuezhi Wang, Dale Schuurmans, Maarten Bosma, Fei Xia, Ed Chi, Quoc V. Le, and Denny Zhou. "Chain-of-thought prompting elicits reasoning in large language models." Advances in Neural Information Processing Systems 35 (2022): 24824-24837.
>
> [3] Kojima, Takeshi, Shixiang Shane Gu, Machel Reid, Yutaka Matsuo, and Yusuke Iwasawa. "Large language models are zero-shot reasoners." Advances in neural information processing systems 35 (2022): 22199-22213.

---

### Official Review · Reviewer_p5Yn · 2023-08-11

**Soundness:** 4

**Excitement:**

4: Strong: This paper deepens the understanding of some phenomenon or lowers the barriers to an existing research direction.

**Paper Topic And Main Contributions:**

The paper proposes a framework to address the weak performance of large language models on complex and multi-step tasks such as math and planning problems. The paper proposes the use of a tree search algorithm to explore possible intermediate steps and actions through roll-outs while relying on the language model itself to act as a world model.

The main contribution of this paper is the reasoning and planning framework, complete with descriptions of the algorithm, reward designs, and methods to adapt the framework to different use cases.

**Questions For The Authors:**

Question A: What was the motivation behind the reward function choices made for different tasks? Is there a reason why the reward system based on the self-evaluation of the LLM is used when evaluating the framework on GSM8k and PrOntoQA benchmarks, but not for the BlockWorld benchmark?

**Reasons To Accept:**

A) The paper proposes a solution that can be applied to most black box language models without the need to fine-tune the model.

B) The paper adapts and demonstrates the use of the proposed framework in the context of a diverse set of problems including planning, math, and logical reasoning tasks.

C) The evaluation method for the framework itself is robust and clearly demonstrates the advantage of using the framework over contemporary baselines, even with the missing reasoning for the choice of reward designs for each task.

**Reasons To Reject:**

~A) The effectiveness of reward designs is not fully explored. The reward systems that use the self-evaluation by the LLM (introduced in line 277) and the likelihood of action (introduced in line 262) can seemingly be applied to any of the tasks considered during evaluation. No analysis is provided regarding whether one design is better than another, or if the systems are task-dependent.~

Update - The author comments provide sufficient justification and experimental results that, when added to the paper, address the above concern effectively.

**Reproducibility:**

5: Could easily reproduce the results.

**Reviewer Confidence:**

4: Quite sure. I tried to check the important points carefully. It's unlikely, though conceivable, that I missed something that should affect my ratings.

**Typos Grammar Style And Presentation Improvements:**

Improvement A: The plan generation example for the GSM8k question in Figure 2 was hard to follow, possibly because the question was abridged. Showing the full question would help readers understand the figure better.

---

> ### Author Rebuttal · Authors · 2023-08-29
>
> We thank the reviewer p5Yn for acknowledging the contribution of our paper to introduce a universal solution for black box language models, adaptable across diverse tasks without fine-tuning, and to demonstrate its effectiveness with robust evaluations. We address the reviewer’s concern about the effectiveness of reward design as follows:
>
> ## The effectiveness of reward designs
>
> Similar to the reward design in RL research, the rewards in LLM reasoning also require some curation and design. In short, we choose the combination of rewards in our current experiments based on specific task heuristics and our exploratory experiments. We generally find that both self-evaluation and action likelihood are beneficial to boost performance but the best combination depends on the nature of the task (with more detailed analyses shown later).
>
> Moreover, as the reviewer suggested, to fully explore the effectiveness of reward designs, we further conduct two ablation studies on various combinations of rewards in Blocksworld and GSM8k datasets as shown in RTable 1 and RTable 2.
>
>
> |Action likelihood|Task-specific|Self-evaluation| Success Rate|
> |-|-|-|-|
> |√|√|×|0.88|
> |√|√|√|0.91|
> |√|×|×|0.46|
> |×|√|×|0.21|
> |×|×|√|0.14|
> |×|×|×|0.02|
>
> **RTable 1: RAP with different reward designs on 4-step problems of Blocksworld** (row 1 is the setting used in the paper)
>
> As shown in RTable 1, we can see the combination of action likelihood and task-specific reward (row 1) can significantly outperform the single reward baselines (row 3, 4, 5). Interestingly, adding the self-evaluation reward can further improve the performance slightly (row 2), indicating the effectiveness of the self-evaluation reward.
>
> |Confidence|Action likelihood|Self-evaluation|RAP$^{(1)}$|RAP$^{(10)}$|+aggr|
> |-|-|-|-|-|-|
> |√|×|×|0.350|0.447|0.490|
> |√|×|√|0.410|0.450|0.503|
> |√|√|×|0.373|0.423|0.443|
>
> **RTable 2: RAP with different reward designs on first 300 examples of GSM8k** (row 2 is the setting used in the paper)
>
> Furthermore, as the results in the first 300 samples of GSM8k (for quick experiments) shown in RTable 2, we can see adding either action likelihood (row 3) or self-evaluation (row 2) on top of confidence reward can boost the RAP$^{(1)}$ (with only 1 iteration) performance of only using confidence reward (row 1), but the self-evaluation rewards lead to the best performance overall. This indicates the importance of self-evaluation reward in guiding reasoning as an effective and computationally efficient prior to exploration.
>
> ## On self-evaluation and likelihood of action
>
> Overall, while we agree with the reviewer that “self-evaluation and the likelihood of action can seemingly be applied to any of the tasks”, their best usage and combination with other rewards require empirical design and understanding of the task nature, and **their effectiveness can vary significantly across different tasks**. Here, we provide some intuitions behind the reward choices:
>
> (a) For the problems in which one reasoning step is short and structured, the action likelihood can be very indicative. Otherwise, it may be disturbed by unimportant tokens and become unreliable. For instance, a single step within the Blocksworld domain typically adheres to specific patterns (e.g., pick/put/stack xxx block…), rendering the action likelihood indicative. However, in the math domain (GSM8k), a reasoning step is expressed in natural language sentences, allowing for greater freedom and potentially introducing noise.
>
> (b) For the problems where it’s easier to recognize some errors afterward than avoid them during generation, self-evaluation emerges as a helpful mechanism for enhancing reasoning accuracy. In mathematical reasoning, LLMs may struggle to generate a correct reasoning step in the first place, but the detection of calculation or logic errors is more feasible. In Blocksworlds, however, assessing the quality of a candidate action is not straightforward and still requires multi-step reasoning. This characteristic diminishes the accuracy of the self-evaluation reward, making it less helpful especially given that likelihood already provides a good intuition for search.
>
> ## Summary
>
> Finally, we will incorporate these additional experiments and discussions in the updated version of our paper. We will also update Figure 2 in the paper to make it easier to follow. We hope our additional experiments on two comprehensive ablation studies will further enhance the soundness of our paper.
>
> At the same time, we invite the reviewer to consider the bigger picture that our paper aims to paint. By introducing a **world model** and a principled **planning** framework, we bridge LLMs, the intuition from cognitive science, and the techniques from reinforcement learning, which might envision a promising path for future autonomous agent design. We believe that this vision makes our paper not just a technical contribution but a catalytic one, opening up a range of new research directions and applications.
> We hope that our responses to the specific concerns could alleviate any reservations about the paper's immediate merits, while we also expect that this broader view will inspire a sense of the work's potential long-term impact. We are eager to include insightful feedback from the reviewers in the revised version of our paper and look forward to contributing to the further advancement of the field.

---

### Meta-Review · Area_Chair_5Lou · 2023-09-19

**Recommendation:** 5

**Metareview:**

This paper presents a novel approach to enhance the reasoning capabilities of large language models using a Reasoning via Planning (RAP) framework, incorporating an external world model and principled planning. The study exhibits a high degree of soundness and provides comprehensive, data-driven support for its claims. Some remaining concerns are over the inherent limitations of LLMs, the design of rewards, as well as a few clarifications needed and a more enriched limitation section needed. The authors’ rebuttal addresses the concerns effectively with additional studies and detailed explanations. Should the paper incorporate all the suggestions, it overall presents an exciting stride in language model research, providing a framework applicable across varying tasks with substantial performance improvements.

---

### Decision · Program_Chairs · 2023-10-07

**Decision:**

Accept-Main

**Comment:**

This paper presents a novel approach to enhance the reasoning capabilities of large language models using a Reasoning via Planning (RAP) framework, incorporating an external world model and principled planning. The study exhibits a high degree of soundness and provides comprehensive, data-driven support for its claims. Some remaining concerns are over the inherent limitations of LLMs, the design of rewards, as well as a few clarifications needed and a more enriched limitation section needed. The authors’ rebuttal addresses the concerns effectively with additional studies and detailed explanations. Should the paper incorporate all the suggestions, it overall presents an exciting stride in language model research, providing a framework applicable across varying tasks with substantial performance improvements.